

# Resistance to starvation of first-stage juveniles of the Caribbean spiny lobster

Alí Espinosa-Magaña[1,2], Enrique Lozano-Álvarez[2] and Patricia Briones-Fourzán[2]

[1] Posgrado en Ciencias del Mar y Limnología, Universidad Nacional Autónoma de México, Ciudad Universitaria, Ciudad de México, México
[2] Unidad Académica de Sistemas Arrecifales, Instituto de Ciencias del Mar y Limnología, Universidad Nacional Autónoma de México, Puerto Morelos, Quintana Roo, México

## ABSTRACT

The non-feeding postlarva (puerulus) of spiny lobsters actively swims from the open ocean to the coastal habitats where it settles and molts to the first-stage juvenile (JI). Because pueruli use much of their energy reserves swimming and preparing for the post-settlement molt, the survival of JIs presumably depends on resuming feeding as soon as possible. To test this hypothesis, the resistance to starvation of JIs of the Caribbean spiny lobster, *Panulirus argus*, was evaluated by measuring their point-of-no-return (PNR, minimum time of initial starvation preventing recovery after later feeding) and point-of-reserve-saturation (PRS, minimum time of initial feeding allowing for food-independent development through the rest of the molting cycle) in a warm and a cold season. Each experiment consisted of eight groups: a continuously fed control (FC) group, a continuously starved control (SC) group, and six groups subjected to differential periods of either initial starvation and subsequent feeding (PNR experiments) or initial feeding and subsequent starvation (PSR experiments). No JIs molted under continuous absence of food (SC). In both PNR experiments (temperature in warm season: $29.79 \pm 0.07\,°C$, mean $\pm$ 95% CI; in cold season: $25.63 \pm 0.12\,°C$) mortality increased sharply after 9 d of initial starvation and intermolt periods increased with period of initial starvation, but were longer in the cold season. The $PNR_{50}$ was longer in the warm season ($12.1 \pm 1.2$ d, mean $\pm$ 95% CI) than in the cold season ($9.5 \pm 2.1$ d). In PRS experiments (temperature in warm season: $29.54 \pm 0.07\,°C$; in cold season: $26.20 \pm 0.12\,°C$), JIs that molted did so near the end of the feeding period; all JIs initially fed for up to 6 d succumbed, and no JIs molted after 13 d of starvation despite having fed previously. The $PRS_{50}$ did not differ between the cold ($13.1 \pm 0.7$ d) and warm seasons ($12.1 \pm 1.1$ d). JIs of *P. argus* exhibit a remarkable resistance to starvation considering that the previous non-feeding, energy-demanding puerulus phase lasts for $\sim$3 weeks. However, JIs appear to have a relatively higher degree of dependence on food to complete development to JII during the cold season than during the warm season. Therefore, JIs of *P. argus* would appear to be more resistant to starvation during the warm season.

Corresponding author
Alí Espinosa-Magaña,
disarm22@hotmail.com

## INTRODUCTION

The Caribbean spiny lobster, *Panulirus argus* (Latreille, 1804) is widely distributed in the wider Caribbean region, where it constitutes one of the most important fishing resources (*Holthuis, 1991*; *Phillips et al., 2013*). In the warm Caribbean waters, *P. argus* breeds throughout most of the year, with a major peak during the spring and a secondary peak in the fall (*Briones et al., 1997*). The larval phase consists of 10 flattened planktonic stages, the phyllosomata (*Goldstein et al., 2008*). Larval development takes place in oceanic waters and lasts 5–9 months, conferring an enormous dispersion potential. The final larval stage undergoes a drastic metamorphosis into the postlarval phase, which consists of a single nektonic stage, the puerulus, which is more similar in shape to an adult lobster but is completely transparent (*Phillips et al., 2006*). Unlike phyllosomata, which are planktotrophic, pueruli do not feed, i.e., they represent a secondary lecithotrophic phase that depends on its own energy reserves (*Kittaka, 1994*; *Lemmens, 1994*; *McWilliam & Phillips, 1997*; *Cox, Jeffs & Davies, 2008*). Pueruli actively swim towards the shore and settle in shallow coastal vegetated habitats, such as seagrass and macroalgal beds (*Butler & Herrnkind, 2000*; *Briones-Fourzán & Lozano-Álvarez, 2001*). A few days after settling, pueruli molt into the first juvenile stage, also known as post-puerulus (e.g., *Limbourn et al., 2008*), which resumes feeding.

Under laboratory conditions, the total duration of the non-feeding puerulus phase of *P. argus* has been 16 to 26 d at 25 °C and 11 to 18 d at 27 °C (*Goldstein et al., 2008*). During this period most of the puerulus energy is allocated to swimming and preparing for the post-settlement molt into the first juvenile stage (JI), which involves further morphological changes (*Lewis, Moore & Babis, 1952*; *Goldstein et al., 2008*; *Ventura et al., 2015*). Swimming for distances of up to tens of kilometers to the settlement site can severely deplete the energy stores of the puerulus, potentially compromising its ability to molt into the JI (*Jeffs, Willmott & Wells, 1999*; *Jeffs, Chiswell & Booth, 2001*). Moreover, resumption of feeding by the JI may depend on local food availability and predation risk (*Lozano-Álvarez, 1996*; *Weiss, Lozano-Álvarez & Briones-Fourzán, 2008*). If the JI cannot restore sufficient energy reserves quickly enough, it could starve to death (*Fitzgibbon, Jeffs & Battaglene, 2014*). Therefore, it is important to determine the resistance to starvation (sensu *Sulkin, 1978*) of JIs of *P. argus*. This information, in addition to increasing knowledge regarding the biology of the early benthic stages of this species, is important for its potential in aquaculture and for the generation of predictive models of local lobster production based on levels of puerulus settlement (*Phillips et al., 2000*).

In the present study, resistance to starvation of JIs of *P. argus* was experimentally determined via two physiological indices, the point-of-no-return (PNR) and the point-of-reserve-saturation (PRS). The PNR is the duration of initial food deprivation that will cause irreversible damage, i.e., that will not allow recovery even after later re-feeding (*Blaxter & Hempel, 1963*; *Anger & Dawirs, 1981*); the PRS is the minimum time of initial feeding after which a later food-independent development to the next stage is possible (*Anger, 1987*; *Anger, 1995*). To our knowledge, the PRS has not been determined for JIs of any spiny lobster species, and the only one for which the PNR of JIs has been determined is *P.*

*cygnus* (*Limbourn et al., 2008*). In the present study, the PNR was determined by subjecting JIs of *P. argus* to a range of initial starvation periods shortly after molting, followed by continuous feeding. The PRS was determined by feeding JIs initially for diverse periods before food was permanently withheld. Each index was estimated during a warm season (summer-autumn) and a cold season (winter-spring) because the metabolic and growth rates of crustaceans in general (e.g., *Anger, 2001*), and of *P. argus* in particular (*Perera et al., 2007*), increase with temperature, potentially affecting the PNR and PRS (*Anger et al., 1981*; *Paschke et al., 2004*; *Bas, Spivak & Anger, 2008*; *Gebauer, Pashke & Anger, 2010*).

## MATERIALS AND METHODS

Given the long larval duration and extremely high rates of mortality of phyllosomata under laboratory conditions (*Goldstein et al., 2008*), our study was conducted using wild postlarvae as in *Limbourn et al. (2008)*. Pueruli of *P. argus* were obtained from a set of 12 artificial seaweed GuSi collectors (*Gutiérrez-Carbonell, Simonín-Díaz & Briones-Fourzán, 1992*) permanently deployed in the reef lagoons of Puerto Morelos (20°52′07″N, 86°52′04″W) and Bahía de la Ascensión (19°49′50″N, 87°27′09″W), Mexico. These collectors are only checked once a month because they are used for long-term monitoring of monthly pueruli settlement. Although monthly settlement in collectors can vary by an order of magnitude within a year, the average catch is 16 individuals per collector per month in Puerto Morelos, and four individuals per collector per month in Bahía de la Ascensión (*Briones-Fourzán, Candela & Lozano-Álvarez, 2008*). Therefore, to increase the number of pueruli, a large mat (1 m × 8 m) of the same artificial seaweed was moored off a dock at Puerto Morelos and checked for pueruli every morning throughout the dark phase of the moon, when settlement levels are typically higher (*Briones-Fourzán, 1994*). Within 1 h of collection, the pueruli were transported in aerated seawater to the laboratory where the experiments were conducted. The necessary permits for pueruli collection were obtained from Comisión Nacional de Acuacultura y Pesca (DGOPA.06695.190612.1737).

The pueruli were individually placed in small covered cylindrical plastic baskets (0.5 l) to allow free water exchange. Each basket was lined with black nylon mesh to prevent escape of the puerulus and to reduce the light level. The baskets were partially submerged and supported by ethylene vinyl acetate ("foamy") floats to provide buoyancy, and were distributed among three tanks (2 m in diameter) with a water level of 0.9 m (approximately 2,800 l each), with 6–7 baskets allotted to each treatment in each tank. The tanks received seawater from an open-flow system. The inflow water was pumped from the Puerto Morelos reef lagoon and passed through a mechanical filter and an ozone chamber (ozone injected at a rate of 0.98 mg/l) to an elevated tank (5,000 l; residence time, 0.3 h), and then to an open reservoir tank (6,300 l; residence time, 3.5 h). Then, the water was pumped through a sand filter (S-244T; Hayward high rate sand filter) to remove suspended particles before its distribution to the experimental tanks. The outflowing water was passed again through an ozone chamber. Water temperature in the tanks was recorded twice a day with an YSI Environmental 556 multi-probe system.

Because each experiment required 160 pueruli (see below) but the number of pueruli that could be collected during any given season (warm or cold) was limited, it was not

possible to conduct more than one experiment per season. Therefore, the PNR experiments were conducted in June–October 2012 (warm season) and January–April 2013 (cold season), and the PRS experiments were conducted in January–April 2014 (cold season) and July–November 2014 (warm season). For each experiment, 160 pueruli were divided into eight groups of 20 individuals each. All individuals were checked twice daily, in the morning and evening. When a puerulus molted into a JI, the exuvia was removed. One group of recently molted JIs was continuously fed (fed control, FC) until all individuals molted into second-stage juveniles (JIIs). Another group remained unfed until all the individuals succumbed (starved control, SC). In the PNR experiments, the six remaining groups of recently molted JIs were subjected to different treatments (periods of starvation in multiples of 3 d) and then fed continuously. Thus, one group was starved for 3 d and then fed (treatment S3), another group was starved for 6 d and then fed (treatment S6), etc. (up to 18 d of initial starvation) (Fig. 1A). In the PRS experiment, the six remaining groups of recently molted JIs were also subjected to different treatments, consisting of feeding periods (also in multiples of 3 d) and then starved. Thus, one group was fed for 3 d and then starved (treatment F3), another group was fed for 6 d and then starved (treatment F6), etc. (up to 18 d of initial feeding) (Fig. 1B). Food, when provided, consisted of a piece of mussel meat (*Perna canaliculus*) changed daily. Mussels are a good food source for early juveniles of *Panulirus* in laboratory conditions, at least during the first four weeks after the molt to JI (e.g., *P. cygnus*: *Glencross et al., 2001*; *P. ornatus*: *Smith, Williams & Irvin, 2005*; *P. interruptus*: *Díaz-Iglesias et al., 2011*). Three-day periods were used to optimize the use of experimental individuals (which are difficult to obtain in large numbers) given that, in the laboratory, the average intermolt period of well-fed JIs of *P. argus* varied between 17 d and 31 d depending on water temperature (*Lellis & Russell, 1990*), whereas the maximal starvation time for JIs of *P. cygnus* from which recovery was observed was 22 d (*Limbourn et al., 2008*). Therefore, JIs of *P. argus* were expected to endure relatively long starvation periods.

For each individual, the first experimental day was one day after molting to JI (*Limbourn et al., 2008*). The influence of the initial starvation periods (in the PNR experiments) or the initial feeding periods (in the PRS experiments) was measured as percent mortality and the average duration of the JI stage of individuals that molted to JII (intermolt period). For the PNR and PRS experiments, duration of stage JI in days (logarithmically transformed to increase homogeneity of variance) was separately subjected to a factorial ANOVA with season and treatment as fixed factors. Starvation tolerance was quantified as the median point-of-no-return ($PNR_{50}$) and point-of-reserve-saturation ($PRS_{50}$). $PNR_{50}$ is the time when 50% of initially starved JIs lost the capability to recover, even after subsequent feeding, and died without molting to JII; $PRS_{50}$ is the time when 50% of initially fed JIs attained the capability to develop through the rest of the molting cycle using internally stored energy reserves (*Anger & Dawirs, 1981*). $PNR_{50}$ and $PRS_{50}$ were estimated by fitting sigmoidal dose–response curves of cumulative mortality to the time of initial starvation or feeding, respectively (*Paschke et al., 2004*; *Gutow et al., 2007*; *Bas, Spivak & Anger, 2008*). Each index was compared between seasons (warm versus cold).

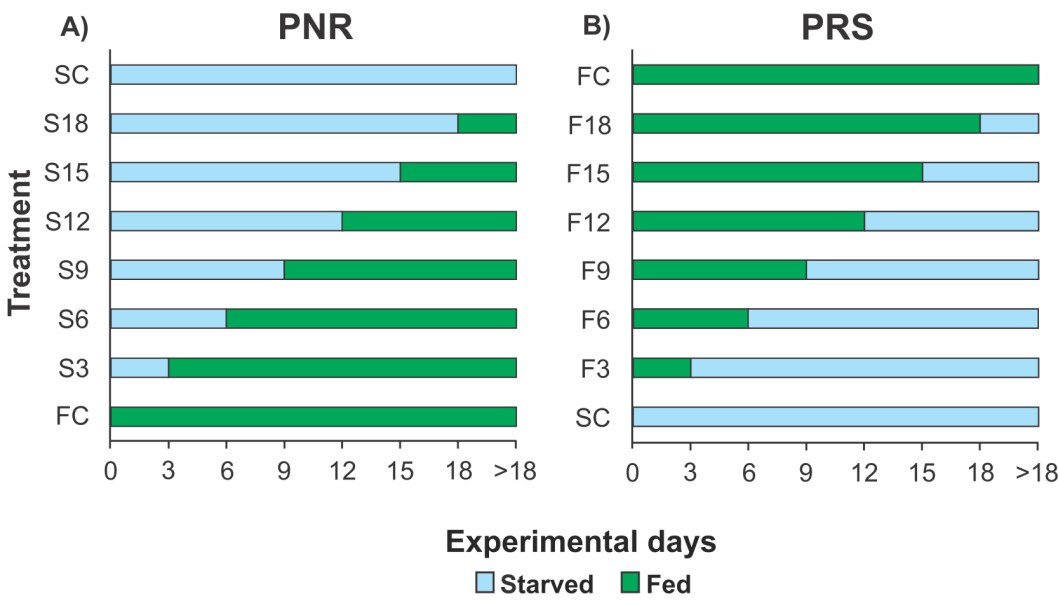

**Figure 1** **Experimental Design of PNR and PRS.** Design of (A) point-of-no-return (PNR; initial starvation, followed by feeding) and (B) point-of-reserve-saturation (PRS; initial feeding, followed by starvation) experiments. FC, continuously fed control; SC, continuously starved control; S3–S18, no. of days of starvation, followed by continuous feeding; F3–F18, no. of days of feeding, followed by continuous starvation. Initial number of *Panulirus argus* pueruli per treatment, FC and SC groups, $n = 20$.

## RESULTS

Throughout the text, results are reported as mean ± 95% CI unless otherwise stated. Statistical results were considered as significant at $p < 0.05$.

### Point-of-no-return (PNR)

In the PNR experiments, water temperature differed significantly between the warm (29.79 ± 0.07 °C) and cold seasons (25.63 ± 0.12 °C) ($t_{176} = 28.65$, $p < 0.001$). The duration of stage JI in continuously fed individuals (i.e., the FC) was longer in the cold season (24.5 ± 3.6 d, $n = 17$) than in the warm season (18.2 ± 1.3 d, $n = 20$) ($t_{35} = 3.44$, $p < 0.001$). In general, the duration of stage JI was significantly affected by season ($F_{1,148} = 68.04$, $p < 0.001$) and treatment ($F_{5,148} = 22.32$, $p < 0.001$) but not by the interaction term ($F_{5,148} = 0.37$, $p = 0.87$); i.e., it increased with the duration of the initial period of starvation in both seasons but was longer for each period in the cold season (Fig. 2). Overall, mortality of JIs increased with number of initial days of starvation, but tended to be higher in a few treatments of the cold season experiment (FC, S3, S6, S12) than in the same treatments of the warm season experiment. The PNR$_{50}$ was 12.1 ± 1.2 d in the warm season and 9.5 ± 2.1 d in the cold season (Fig. 3), a significant difference ($F_{1,12} = 7.924$, $p = 0.015$).

### Point-of-reserve-saturation (PRS$_{50}$)

In the PRS experiments, water temperature differed significantly between the cold season (26.20 ± 0.12 °C) and the warm season (29.54 ± 0.07 °C) ($t_{285} = 24.23$, $p < 0.001$). The
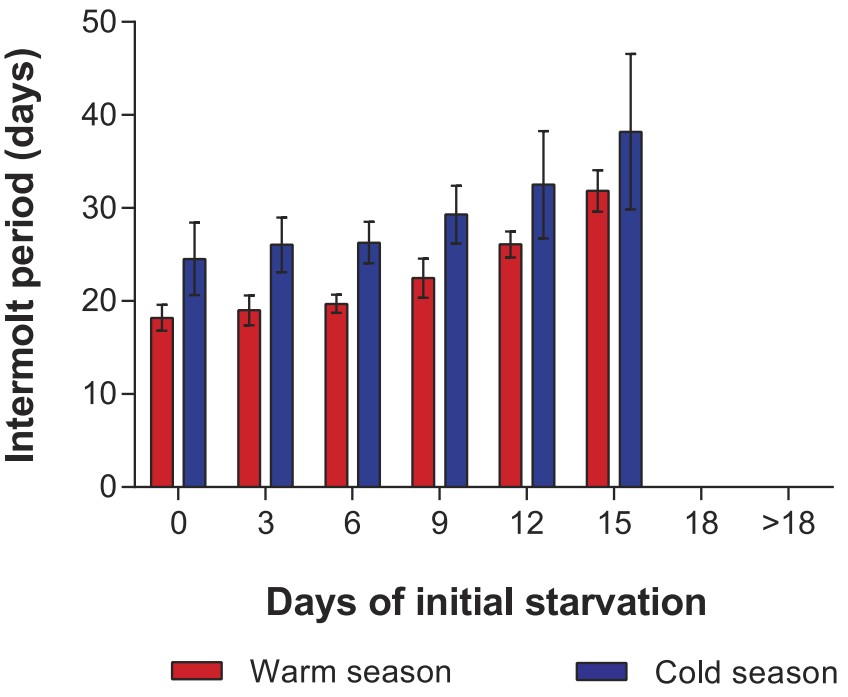

**Figure 2 Intermolt period in PNR experiments.** Intermolt period (days) between JI and JII stages of *Panulirus argus* subjected to different periods of starvation followed by continuous feeding in experiments to determine the point-of-no-return. Red columns, warm season, June–October 2012; blue columns, cold season, January–April 2013. Results from treatments with two survivors or fewer were omitted. Error bars, 95% CI.

duration of stage JI in individuals from the FC was significantly longer in the cold season ($25.7 \pm 2.3$ d, $n = 20$) than in the warm season ($16.3 \pm 1.3$ d, $n = 20$) ($t_{38} = 7.05, p < 0.001$). Overall, the duration of stage JI was significantly affected by season ($F_{1,119} = 37.9, p < 0.001$) and treatment ($F_{4,119} = 8.71, p < 0.001$), but the interaction term was also significant ($F_{4,119} = 3.76, p < 0.001$), indicating that the intermolt period in the different treatments did not follow the same pattern in both seasons (Fig. 4). In particular, the intermolt period was significantly longer in treatments F15 and F18 and in the FC of the cold season than in the same treatments of the warm season (Fig. 4).

Mortality of JIs decreased with increasing period of initial feeding, but was higher in almost all treatments of the cold season than in those of the warm season. In both seasons, all individuals in the SC and in treatments F3 and F6 succumbed. In both experiments, with the exception of the fed control group, most JIs molted near the end of the initial feeding period, with a maximum of 13 d between the end of the initial feeding period and molting. The PRS$_{50}$ was $13.1 \pm 0.7$ d in the cold season and $12.1 \pm 1.1$ d in the warm season (Fig. 5). These values did not differ significantly ($F_{1,12} = 3.603, p = 0.082$).

## DISCUSSION

Temperature has an important effect on the physiology and ecology of ectotherms such as spiny lobsters. For example, in Florida, USA, *Witham (1973)* reported that small juveniles

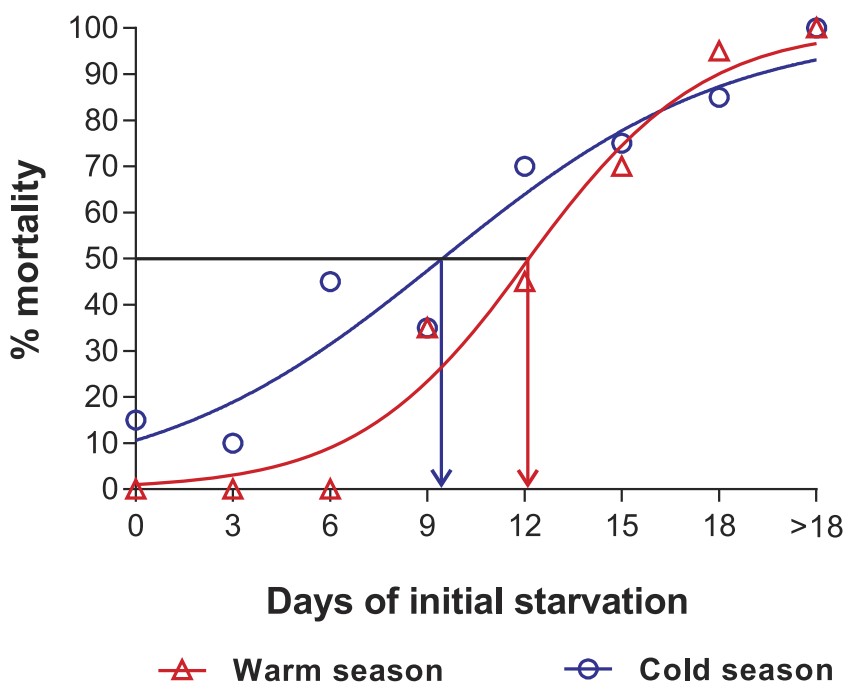

**Figure 3 Sigmoidal dose-response curves—PNR experiments.** Sigmoidal dose-response curves fitted to the mortality (%) of JIs of *Panulirus argus* subjected to a period of starvation before being fed. Triangles, warm season, June–October 2012; circles, cold season, January–April 2013. PNR$_{50}$ values: 12.1 $\pm$ 1.2 d in the warm season and 9.5 $\pm$ 2.1 d in the cold season.

of *Panulirus argus* were intolerant of sustained temperatures below 15.6 °C or above 30.0 °C, and that lobsters held below 20 °C exhibited relatively little growth. In laboratory experiments held in Cuba, oxygen consumption and metabolism of early benthic and older juveniles of *P. argus* increased with temperature within the range of 19 to 30 °C (*Brito et al., 1991*; *Perera et al., 2007*). *Lellis & Russell (1990)* examined growth and survival of postpueruli and early juveniles of *P. argus* at different temperatures and found that these stages grew faster at 30 °C due to shorter intermolt periods and greater size increments than at 24, 27, or 33 °C. In our study, mean water temperature during the PNR and PRS experiments was 25.9 and 26.2 °C in the cold season, respectively, and 29.8 and 29.5 °C in the warm season, respectively. In both types of experiment the duration of stage JI of continuously fed individuals (FC) was significantly longer in the cold season than in the warm season.

*Limbourn et al. (2008)* estimated the PNR$_{50}$ of JIs of *P. cygnus*, a subtropical species, at 22 d, which is about twice the PNR$_{50}$ estimated for the tropical *P. argus* in the present study. Regardless, the resistance to starvation of JIs of *P. argus* is remarkable, considering the energetic demand imposed on the non-feeding pueruli during their transit from oceanic waters to the coast (e.g., *Fitzgibbon, Jeffs & Battaglene, 2014*). As previously found by *Limbourn et al. (2008)* for *P. cygnus*, no JIs of *P. argus* were able to molt in complete absence of food (SC). Dependence on exogenous food of JIs to complete development has also been found in other decapods, even in some species with facultative lecithotrophic postlarvae (*Calado et al., 2010*). An exception is the red cherry shrimp, *Neocaridina davidi*,

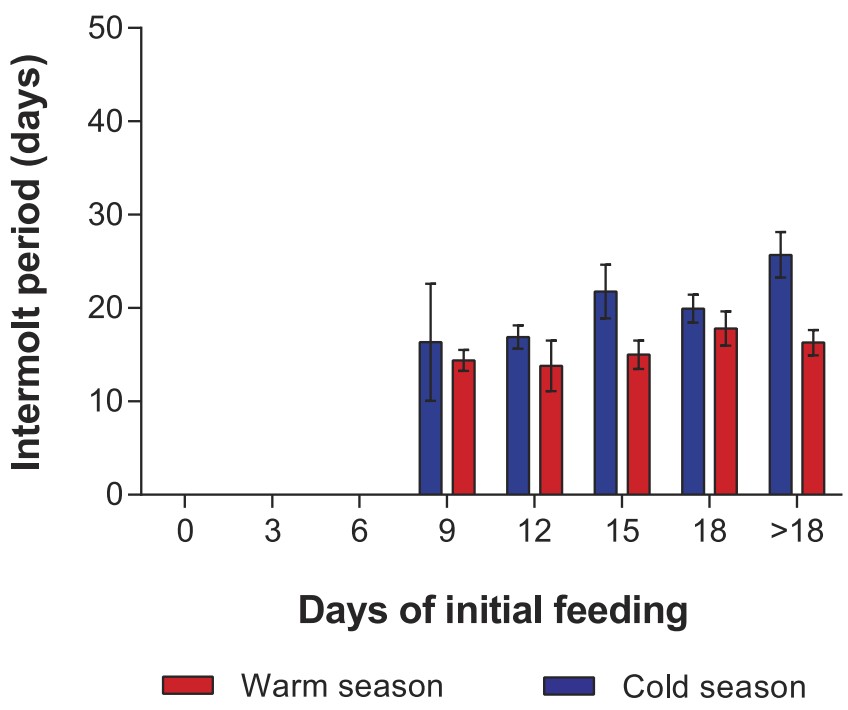

**Figure 4** **Intermolt period in PRS experiments.** Intermolt period (days) between JI and JIII stages of *Panulirus argus* subjected to different periods of feeding followed by continuous starvation in experiments to determine the point-of-reserve-saturation. Blue columns, cold season, January–April 2014; red columns, warm season, July–November 2014. Results from treatments with two survivors or fewer were omitted. Error bars, 95% CI.

in which JIs were able to molt even in complete absence of food, probably because this species has a much abbreviated post-hatching development (*Pantaleão et al., 2015*). In our two PNR experiments, the duration of stage JI increased with increasing period of initial starvation. A delay in molting caused by lack of food has been documented in early life stages of other marine and freshwater crustacean species (e.g., *Anger & Dawirs, 1981*; *Mikami, Greenwood & Gillespie, 1995*; *Abrunhosa & Kittaka, 1997*; *Stumpf et al., 2010*; *Calvo et al., 2012*) except *N. davidi*, in which the duration of stages JI and JIII was not affected by the duration of initial starvation periods (*Pantaleão et al., 2015*).

In the PRS experiments, the few individuals that molted to JII in treatments F9 and F12 did so near the end of the feeding period, and no individuals molted after 13 d of starvation despite having been previously fed, resulting in an apparent increase in duration of stage JI with duration of initial feeding period. This result contrasts with the results of the PNR experiments, in which seven individuals from treatments S15 and S18 of the warm season experiment and eight from treatments S15 and S18 of the cold season experiment were able to molt. The higher tolerance to starvation of individuals from the PNR experiments may be due to a suspension of the molting cycle, i.e., an arrest of development in the intermolt stage (stage C in Drach's molt classification system) of individuals subjected to initial starvation, with a subsequent shift to premolt stage (stage D) with further feeding (*Anger, 1987*). However, in PRS experiments, if individuals reach the premolt stage during the

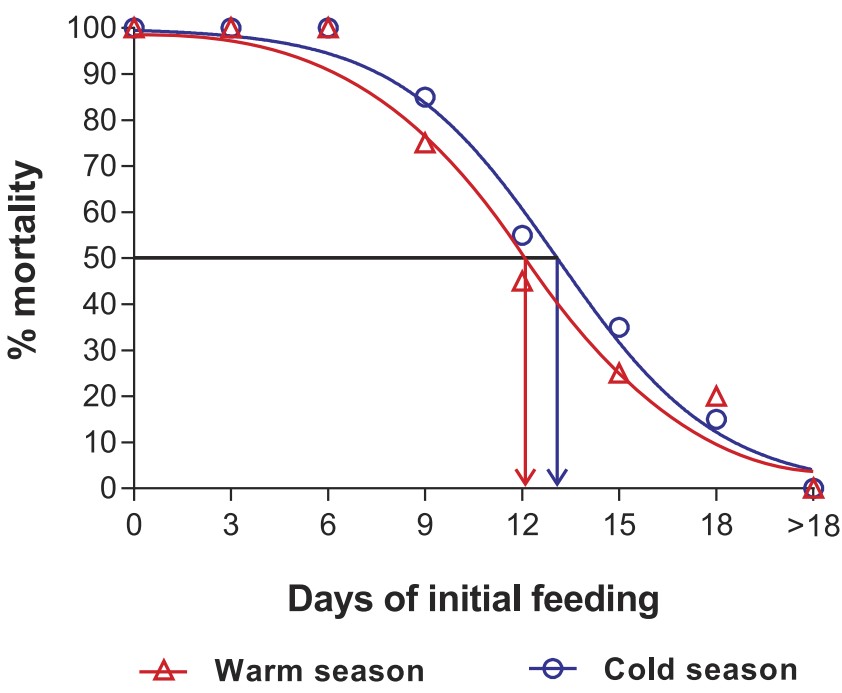

**Figure 5  Sigmoidal dose-response curves—PRS experiments.** Sigmoidal dose-response curves fitted to the mortality (%) of JIs of *Panulirus argus* allowed periods of feeding before being starved. Circles, cold season, January–April 2014; triangles, warm season, July–November 2014. $PRS_{50}$ values: $13.1 \pm 0.7$ d in the cold season and $12.1 \pm 1.1$ d in the warm season.

period of initial feeding and then are subjected to continuous starvation, an unsuccessful attempt to molt will lead to death because an arrest of development in premolt is not possible (*Anger, 2001*).

The PRS occurs during the transition between late intermolt (stage $C_4$) and early premolt (stage $D_0$), which is also known as the "$D_0$ threshold," a critical period for crustaceans in terms of nutritional requirements (*Anger, 1987*; *Anger, 2001*). The $D_0$ threshold has been confirmed in early life stages of decapod crustaceans from different families, e.g., Crangonidae (*Paschke et al., 2004*), Diogenidae (*Harms, 1992*), Majidae (*Figueiredo et al., 2008*; *Guerao et al., 2012*), and Portunidae (*Harms et al., 1990*). In our study, results of both PRS experiments, with high mortality rates in individuals subjected to treatments F3, F6 and F9, and shorter intermolt periods and higher mortality rates than in PNR experiments, as well as the similarity in $PRS_{50}$ values between seasons (unlike in $PNR_{50}$ values), suggest that feeding in *P. argus* JIs is more important near the PRS (or $D_0$ threshold) than immediately after molting to the first-stage juvenile.

Although $PRS_{50}$ values did not differ with season, the ratio of the PRS to the total duration of stage JI in individuals from the FC was higher during the warm season (74%) than in the cold season (51%). Therefore, despite the significantly longer duration of stage JI in the cold season, the PRS was reached earlier than in the warm season. A similar pattern was observed in the zoea I of the shrimp *Crangon crangon*, in which the ratio of the PRS to the stage duration was 32% in the summer and 23% in the winter (*Paschke et al., 2004*).

However, the PRS represents about one-third of the total intermolt time in the early phases of many crustaceans, including *C. crangon*, the zoeae I and II of the spider crab *Hyas araneus* (*Anger & Dawirs, 1981*), the zoeae I of the intertidal crab *Neohelice granulata* (*Bas, Spivak & Anger, 2008*), the phyllosoma I of *Panulirus cygnus* (*Liddy, Phillips & Maguire, 2003*), and stage JIII of the crayfish *Cherax quadricarinatus* (*Stumpf et al., 2010*).

The ratio of the PNR to the total duration of the JI stage of individuals from the FC was 66% in the warm season and 39% in the cold season. Therefore, JIs of *P. argus* would appear to be more resistant to starvation during the warm season. This result, although consistent with the optimum temperature of ∼30 °C for development of this stage (*Lellis & Russell, 1990*), may seem counterintuitive given that metabolic and growth rates increase with temperature (*Anger, 2001*). However, a longer $PNR_{50}$ during the warm season may reflect a reduction in the metabolic responses of JIs as a compensatory physiological mechanism, as has been documented in other crustaceans (e.g., *Litopenaeus setiferus*: *Sánchez et al., 2002*). Alternatively, it may reflect an enhanced ability of JIs to sequester and store reserves at higher temperatures, as suggested by *Smith, Kenway & Hall (2010)* for phyllosomata of tropical spiny lobsters (*P. ornatus* and *P. homarus*), which exhibited a greater tolerance to starvation (longer $PNR_{50}$) than phyllosomata of temperate spiny lobsters.

Despite the commercial importance of *P. argus*, the present study is the first to address resistance to starvation in the early benthic juveniles of this species after they have undergone a protracted lecithotrophic stage that may lead to their return to coastal waters from distances of up to tens of kilometers offshore. However, as *P. argus* juveniles tolerate a wider temperature range across their geographic distribution than those recorded in the present study, further studies subjecting JIs to different experimental temperatures are needed to examine how more extreme temperatures would affect the PNR and PRS of this tropical species. Also importantly, measures of reserve substances utilization and storage by both fed and starved JIs should be considered in future studies (e.g., *Simon et al., 2015*).

## ACKNOWLEDGEMENTS

The authors wish to acknowledge the invaluable technical assistance provided by F Negrete-Soto and C Barradas-Ortiz in the set-up of experiments and throughout the study. JP Huchin-Mian, R Martínez-Calderón, R Candia-Zulbarán, R Muñoz de Cote-Hernández, L Cid-González, and H Canizales-Flores provided additional help. We also thank Ann Grant for her assistance in reviewing the English.

### Funding

This study was funded by Universidad Nacional Autónoma de México and Consejo Nacional de Ciencia y Tecnología (Project 101200-Q and a PhD scholarship for AEM). The funders had no role in study design, data collection and analysis, decision to publish, or preparation of the manuscript.

### Grant Disclosures

The following grant information was disclosed by the authors:

Universidad Nacional Autónoma de México.

Consejo Nacional de Ciencia y Tecnología: 101200-Q.

### Competing Interests

The authors declare there are no competing interests.

### Author Contributions

- Alí Espinosa-Magaña conceived and designed the experiments, performed the experiments, analyzed the data, wrote the paper, prepared figures and/or tables, reviewed drafts of the paper.
- Enrique Lozano-Álvarez conceived and designed the experiments, contributed reagents/materials/analysis tools, wrote the paper, reviewed drafts of the paper.
- Patricia Briones-Fourzán conceived and designed the experiments, analyzed the data, contributed reagents/materials/analysis tools, wrote the paper, reviewed drafts of the paper.

### Field Study Permissions

The following information was supplied relating to field study approvals (i.e., approving body and any reference numbers):

Comisión Nacional de Acuacultura y Pesca.

Approval number: DGOPA.06695.190612.1737.

### Data Availability

The raw data has been supplied as a Supplementary File.

### Supplemental Information

Supplemental information for this article can be found online at http://dx.doi.org/10.7717/peerj.2852#supplemental-information.

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
