# Peer review of "Resistance to starvation of first-stage juveniles of the Caribbean spiny lobster"

_PeerJ, doi:10.7717/peerj.2852_

## Round 0.1 · original submission · Minor Revisions

I have heard back from three reviewers, all of whom feel your work needs some editing and changes before being acceptable for publication. In particular, reviewers 1 and 3 offer some concrete comments, and I would like to see detailed responses to their points.

·

Basic reporting

No comments.

Experimental design

No problem here but I've noted some concern in trying to extrapolate the results too far. This relates to experimental design because comparison between seasons was a reasonable design for testing the effect of season but wasn't ideal to test effect of temperature. So need to be careful in making comments such as the results have implications for understanding effects of climate change.
I'd note that results were very clear which indicates good design overall, good sample size etc.

Validity of the findings

No comment.

Additional comments

1. L55 “However, in various parts of the Caribbean, including Mexico, catches of P. argus have suffered a substantial decline since the late 1990s (Aguilar et al., 2003; Ehrhardt, Puga & Butler, 2011)”.
Consider redoing this sentence. FAO data for all landings of the species doesn’t show a substantial decline. Appears more like catches have been stable since 1990s or perhaps a slight decline. It’s probably true that catches have declined substantially in some parts of the Caribbean but if overall catches are stable, then it’s probably also true that other areas have had increases. Catches are not necessarily a good indicator of abundance anyway, which is the point of this sentence. I think there’s a need for research on juvenile stages regardless of trends in recruitment so this sentence isn’t really necessary anyway.
http://www.fao.org/fishery/species/3445/en
2. Discussion of temperature needs to make the point that this is an extremely stable system. Temperature does vary between the warm and cold seasons but barely. The fact that any effect was detected with such a small range is surprising for me at least. Presumably P. argus are exposed to much higher / lower temperature in other parts of their range so the temperature effect reported here would be that more elsewhere.
3. L 221. “consequently” doesn't make sense in this flow of text.
4. L274. Sentence seems to be missing words. Isn’t clear.
5. L277. I disagree that the results have “implications for (understanding effects of?) climate change”. Juveniles were exposed to only a very narrow range of temperatures between seasons and the experiment was not structured to consider extremes beyond those encountered during the experiments. There’s nothing in the method that tells us whether either measure of starvation would alter with climate changes.
6. I’d be tempted to remove the last sentence of the discussion. These justifications for the research (eg aquaculture) seem to be drawing a long bow. Aquaculture operations aren’t currently commercial and if they were I think it’s safe to assume they’ll supply food to satiation and not include starvation in the management. This PeerJ journal is happy to publish research for the sake of improving knowledge so it doesn't need these sort of justifications that other journals seem to want (for some reason).

Reviewer 2 ·

Basic reporting

- The manuscript could be written with more clarity, it seems a little ambiguous specially in abstract material and methods section.
- The introduction and background is sufficient and relevant.
- The manuscript is well structured
- Figures are relevant
- Results relevant to the hypothesis are presented
- The appropriate data are available but should be more organised

Experimental design

- The submission describe very important original primary research
- research question should be more clear
- The investigation have been conducted rigorously and to a high technical standard
- Method section is a little lengthy and would better to shorten
- The research have been conducted in conformity with the prevailing ethical standards in the field.

Validity of the findings

- The data are robust and controlled, the statistical method are correct.
- The conclusion is better to be stated separately

Additional comments

comments are stated in manuscript word text file
- Please make abstract clear and brief specially in lines 30-37
- Lines 93-100 would be better to transfer to material and methods or write in proper way as goal of the study
- How did you eliminate the effect of biological variation of postlarvae (e.g, genetic differences, difference in thermal preference or starvation resistance) in your experiment

Reviewer 3 ·

Basic reporting

In general, the manuscript is well written and presented. It clearly identifies its aims and the purpose of the study, that being the PNR or PRS of a novel species or life stage of spiny lobster. PNR and PRS analysis is a well-established techniques and has been long examined in crustacean biology. In that sense this study has limited novelty other than for another species of life stage that has been overlooked. However, considering the likely high biological importance of re-feeding of the post-puerulus, this study does provide an appropriate contribution to the field. The figures in the review draft were too small but I assume this will be corrected in the published manuscript. The article referenced an appropriate level of published papers however should consider the revision of “Smith, G., Kenway, M., Hall, M., 2010. Starvation and recovery ability of phyllosoma of the tropical spiny lobsters Panulirus ornatus and P. homarus in captivity Journal of the Marine Biological Association of India 52, 249.”

Experimental design

As previously mentioned PNR or PRS analysis is a well-established techniques and the authors have done a good job in their experimental design. I am particularly impressed with the high number of experimental replicates. Method are generally well described however could benefit from some more detail about culture systems. Particularly the level of ozonation needs to be better defined as in appropriate seawater ozonation can have dramatic influence on the development, moulting and survival of crustaceans.

Validity of the findings

The discussion and conclusions of the study are generally sound. However, I question if the authors have sufficiently discussed possibly the most profound finding of the study; that starvation resistance is improved in warmer temperatures. This finding is counter intuitive considering the strong effect of temperature on metabolic rates and consequently energy use. The study would have greatly benefited by more advance measures of lobster energy utilisation and storage including metabolic rate and carcass biochemical/energy composition which should be considered in future studies. In the absence of these measurement I suggest that the authors should better speculate on the physiological basis of improved starvation resistance at warm temperature. This could include an assessment of total energy use at differing temperatures using published metabolic and Q10 values in relation to intermoult duration. Authors should also review other published work on the effect of starvation and refeeding on lobster metabolism and energy storage, for example see: Simon, C.J., Fitzgibbon, Q.P., Battison, A., Carter, C.G., Battaglene, S.C., 2015. Bioenergetics of nutrient reserves and metabolism in spiny lobster juveniles Sagmariasus verreauxi: Predicting nutritional condition from hemolymph biochemistry. Physiological and Biochemical Zoology 88, 266-283.

---

## Round 0.2 · accepted · Accept

The manuscript has been well revised, and is now acceptable for publication. Please note two small changes in the attached PDF, which should be done at the proof stage if not earlier.

I look forward to seeing the published version of this work!